# Clinical and Radiological Outcome of Posterior Cervical Fusion Using Philips AlluraXper FD20 Angiography Suite

**DOI:** 10.3390/brainsci15020160

**Published:** 2025-02-06

**Authors:** Armando Dolp, Abdussalam Khamis, Javier Fandino, Jenny C. Kienzler

**Affiliations:** 1Department of Neurosurgery, Cantonal Hospital Aarau, 5001 Aarau, Switzerland; armando.dolp@me.com; 2Department of Spine Surgery, Hospital Center Biel, 2501 Biel, Switzerland; khamis_med@yahoo.com; 3Department of Neurosurgery, Hirslanden Medical Center, 5001 Aarau, Switzerland; javier.fandino@hirslanden.ch; 4Department of Neurosurgery, Hirslanden Medical Center, 8032 Zurich, Switzerland; 5Department of Neurosurgery, University Hospital Lausanne, 1005 Lausanne, Switzerland

**Keywords:** cervical spine, fusion, posterior, surgical technique, outcome, hybrid OR, intraoperative CT

## Abstract

Background: Posterior cervical fusion (PCF) is widely used for cervical spinal cord decompression with/without fusion. In our hybrid operating room, intraoperative computed tomography (iCT) is routinely used to verify screw placement. This study analyzed clinical and radiological outcomes after PCF and evaluated iCT benefits for detecting screw misplacement. Methods: Nineteen patients underwent PCF between March 2012 and April 2016 for degenerative (n = 6), neoplastic (n = 7), and traumatic (n = 6) conditions. Seven patients had primary PCF, while twelve underwent PCF following anterior fusion due to segmental instability with cervical malalignment (n = 11) or tumor progression (n = 1). Results: The mean patient age was 59 ± 11 years, with 63% male patients. The median follow-up was 21 months. PCF averaged 4.74 segments (range: 1–9). At follow-up, 79% reported pain improvement and normal sensorimotor function. Of six patients with preoperative paresis, five showed improved muscle strength. No persistent gait disturbances occurred. Complications requiring revision occurred in four patients (21%): three surgical site infections and one cerebrospinal fluid leak. One perioperative death occurred (5%). iCT detected incorrect screw placement in seven patients (36%), allowing the immediate repositioning of eight screws, preventing later revision surgeries. The overall fusion rate was 92%. Conclusions: PCF with iCT is safe and effective for various cervical spine pathologies, yielding good long-term clinical outcomes. iCT effectively detects and enables immediate correction of screw malposition, reducing revision surgery needs. This imaging modality demonstrates high sensitivity and specificity for identifying clinically relevant screw malpositions.

## 1. Introduction

A wide range of cervical spine disorders can be managed with various surgical treatments. These disorders include degenerative and neoplastic pathologies affecting the three columns of the spine, which can lead to micro or macro instability, resulting in the compression of the nerve roots or spinal cord. In cases of spinal cord injury accompanied by progressive neurological deficits, early decompression surgery performed within 24 h of symptom onset has been associated with improved neurological outcomes [1]. However, the optimal timing of surgical intervention for cervical spondylotic myelopathy remains more controversial. While some studies [2] have reported better short-term relief of neck pain following surgery compared to conservative management, long-term benefits have not been consistently demonstrated [3]. As a result, surgical intervention is typically recommended only for patients presenting with moderate to severe signs of myelopathy or progressive neurological deficits [4].

Several surgical techniques have been described for the decompression of the cervical spine, including both anterior and posterior approaches, with or without fusion. Among the well-established anterior techniques are anterior cervical discectomy and fusion (ACDF) and anterior cervical corpectomy and fusion (ACCF) [5,6,7,8]. The use of an anterior plate can provide additional stability, aiming to enhance fusion rates and prevent pseudoarthrosis [9]. Alternative posterior approaches include laminoplasty, laminotomy, and laminectomy, which may be combined with fusion of the affected segments. However, surgical procedures involving screw placement carry an inherent risk of misplacement, potentially leading to neurological or vascular injuries and necessitating revision surgery.

Since 2010, our institution has used a hybrid operating room equipped with intraoperative computed tomography (iCT) to confirm the correct placement of screws and minimize the need for revision surgery. In a previous study, we reported the high sensitivity and specificity of iCT in identifying incorrect pedicle screw placement during lumbar spinal instrumentation [10].

While ACCF and ACDF have remained the standard techniques for decompression and fusion of the cervical spine at our institution, the posterior approach has recently gained more frequent use in the management of degenerative and neoplastic pathologies, as well as cervical spine injuries. The current study aims to evaluate the accuracy of iCT in posterior cervical spine fusion and to report the associated clinical and radiological outcomes.

## 2. Materials and Methods

This study included patients who underwent posterior cervical fusion in our hybrid operating room between March 2012 and April 2016. Lateral mass screws were used for levels C3 to C6, while transpedicular screws were preferred for C7 and the thoracic spine. One patient underwent an isolated C1–C2 fusion using the Harms/Goel technique, and another patient received a fusion from the occiput to C5 combined with an atlantoaxial fusion as described by Magerl.

The following patient characteristics were recorded: age at time of surgery, patient history, neurological status, previous cervical spine surgeries including the approach, adverse events, and revision surgeries during the follow-up period. Postoperative immobilization of the cervical spine was achieved using either a stiff cervical collar or a halo vest.

Physicians from our department conducted neurological examinations at the initial presentation, discharge, and last follow-up. The clinical assessment included level of neck pain and radicular pain using the visual analog scale (VAS [11]), as well as assessing sensorimotor deficits, gait disturbance, and deep tendon reflexes. For patients who were still alive, an additional long-term follow-up assessment was performed in the outpatient clinic. In these patients, we additionally collected the Neck Disability Index (NDI) [12], the Nurick Scale [13], the modified Macnab Criteria [14], and Odom’s Criteria [15].

Routine imaging, including anterior–posterior, lateral, and flexion–extension X-rays, computed tomography (CT), and magnetic resonance imaging (MRI), was performed before and after surgery. The first authors and an independent radiologist analyzed these images. The radiological endpoints included the fusion rate using the Lenke and Bridwell fusion classification [16], the number of misplaced screws, and any loosening or fractures of the screws. Additionally, changes in alignment and signs of myelopathy were assessed.

Before 2010, both transpedicular and lateral mass screw placement was guided by conventional two-directional X-rays (lateral and anterior–posterior). Although postoperative CT scans were routinely performed during this period, the rate of misplaced screws was not systematically documented.

In 2010, the Cantonal Hospital Aarau installed a hybrid operating room equipped with a Philips AlluraXper FD20 (Philips Healthcare, Best, The Netherlands) angiography suite featuring cone beam CT with a rotating C-arm. This system was chosen for its versatility in both diagnostic imaging and interventional endovascular procedures. For spinal screw placement, the C-arm rotates around the target area to generate a three-dimensional scan, enabling immediate verification of screw positioning. All patients in this study underwent intraoperative CT scanning. When significant screw misplacement was detected, immediate repositioning was performed until correct placement was confirmed by final intraoperative CT. As a standard practice at our institution, patients underwent a postoperative CT scan with thin-slice reconstructions of 0.5 mm in the axial plane and 3 mm in the sagittal/coronal planes to rule out any relevant screw misplacement. This is performed due to safety considerations and quality control, as the postoperative CT scan provides better imaging quality compared to the iCT. The postoperative CT scans were then used to validate the iCT findings and to calculate specificity, sensitivity, and accuracy.

The intraoperative CT scan software does not allow us to measure distances in millimeters with precise accuracy. As a result, we defined four categories to classify screw placement. A screw was considered to have correct positioning if it was entirely surrounded by the substance of the pedicle or lateral mass. If a screw extended beyond the cortical bone of the pedicle or lateral mass but did not protrude more than one-third of its diameter, it was recorded as a minor violation. Screws that bulged from the bony borders by more than one-third of their diameter but did not exceed the full diameter were classified as moderate violations. Any screw protrusion exceeding the full diameter was labeled as a severe violation.

Neuromonitoring with somatosensory evoked potentials (SSEPs) and motor evoked potentials (MEPs) is not routinely performed during spinal fusion procedures at our institution. However, these monitoring techniques are regularly employed during the treatment of intradural pathologies.

### Statistical Analysis

We present the results using percentages, means with standard deviations, and medians for various parameters, including patient characteristics, extent of spinal fusion, symptoms at discharge, follow-up and long-term outcomes, radiological findings, complications, and adverse events. Additionally, we evaluated the sensitivity (percentage of detected condition), specificity (percentage of detected absence of condition), and accuracy (percentage of overall correct assignment) of the iCT scan in assessing screw placement, using the postoperative CT scan as the reference.

## 3. Results

### 3.1. Patient Characteristics

Between March 2012 and April 2016, a total of 19 patients, consisting of 12 males (63%) and 7 females (37%), underwent posterior cervical fusion at our institution. The mean age at the time of surgery was 59 (±11) years.

Among these patients, seven (37%) had a neoplastic mass in the vertebral column, while six patients (32%) had degenerative changes as the indication for surgery. The remaining six patients (32%) had suffered a traumatic cervical spine injury with fracture. Despite the wide variation in the indications for cervical posterior fusion within our series, the different entities demonstrated similar results. In most cases, neck pain was the primary indication for surgery (Table 1).

### 3.2. Extent of Surgical Spinal Fusion

Prior to the posterior cervical fusion, 12 patients (63%) had already undergone an anterior approach for cervical fusion. For seven patients (37%), posterior spinal fusion was the primary treatment option. The cervical fusion procedures involved a range of one to nine segments, with a mean of 4.74 segments. In six cases (32%), the fusion was confined to the cervical spine. However, in five additional posterior fusions (26%), it was necessary to include the occipito-cervical junction. Furthermore, in 11 cases (58%), the fusion extended to the upper thoracic spine. One patient (5%) developed kyphosis following a chordoma resection via an anterior approach (ACCF) and required a fusion spanning nine segments from the occipito-cervical junction to Th2 (Figure 1 and Table 2).

Following the surgery, patients were immobilized with a stiff cervical collar until the first follow-up, which occurred 4 to 6 weeks postoperatively. In three cases involving fusion to the upper thoracic spine, a corset was prescribed. One patient (5%) who underwent fusion from the occipito-cervical junction to C6 required a halo vest for immobilization.

### 3.3. Symptoms at Discharge

At the time of initial presentation, 16 patients (84%) reported neck pain, while 4 patients (21%) experienced radicular pain. Upon clinical examination, sensory deficits were found in seven patients (37%), and paresis was present in six cases (32%). Gait disturbance and hyperreflexia were each observed in two patients (11%). At discharge, 15 patients (94%) in the neck pain group demonstrated improvement in their pain levels, with 1 patient becoming completely pain-free. One patient (6%) showed no change compared to their preoperative pain level. No new postoperative neck pain was reported at discharge. All patients with radicular pain experienced improvement, with one patient in this group becoming pain-free after surgery. Motor deficits improved in three out of six patients (50%), with one patient having no residual paresis at discharge. Muscle strength remained stable in the other three cases (50%). Sensory disturbances improved in four out of seven patients (57%), with two patients (29%) recovering normal sensibility and three patients (43%) remaining unchanged. Regarding myelopathy symptoms, both patients (100%) in the gait disturbance group had no residual symptoms at discharge. As anticipated, hyperreflexia did not normalize during the brief period until discharge.

### 3.4. Follow-Up and Clinical Long-Term Outcome

Among the 19 patients in this study, 15 had a clinical follow-up of at least 12 months, with a median follow-up duration of 21 months after surgery. One patient died 22 days after the surgery, and no follow-up data could be recorded after discharge. At the last follow-up, 11 out of 15 patients (73%) who had complained of pain before surgery reported an improvement in their overall pain level, including both radicular and neck pain. Three patients (20%) reported no changes, and one patient (7%) experienced an increase in neck pain compared to their preoperative state. Neurological examination revealed that 15 patients (83%) had no sensorimotor deficits. An increase in muscular strength was recorded in four out of five patients (80%), with two patients (33%) achieving normal strength. One patient (20%) remained stable. Among the six patients in the sensory deficit group, only one patient (17%) had a residual sensory deficit with unchanged symptoms. Neither of the two patients with preoperative gait impairment showed residual gait disturbance at the last follow-up (Figure 2A,B and Figure 3).

Seven patients (37%) were prospectively followed up in our outpatient clinic. Their Neck Disability Index Score (NDI) ranged from 0 to 24 points. However, six patients (86%) showed only mild or no disability (NDI < 15). The corresponding median for NDI was 4. The Nurick Scale was assessed in two patients with myelopathy symptoms and gait disturbance. The median Nurick Scale score was 0. According to the modified Macnab criteria, a good or excellent outcome was achieved in five patients (71%). In terms of Odom’s criteria, five patients (71%) reported a good or excellent outcome. There was no poor outcome in our current series.

### 3.5. Radiological Findings

Long-term radiological follow-up was not routinely conducted, with a median radiological follow-up of 12 months. Thirteen out of nineteen patients (68%) underwent a CT scan of the fused segments at least 12 months after the surgery. In this group, the fusion rate was evaluated using the Lenke and Bridwell classification system. A solid fusion across the entire fused area was observed in nine patients (69%). Among these, five cases (38%) achieved trabeculated bilateral fusion (Lenke and Bridwell Grade A), while four cases (31%) had unilateral fusion (Grade B). Small, non-solid fusion masses (Grade C) were observed in three patients (23%). One patient (8%) had a non-union in one or more of the fused segments (Grade D) (Table 3).

During the radiological follow-up, screw loosening was observed in five patients (38%), involving a total of 17 screws. However, due to the absence of symptoms, no revision surgery for screw replacement was necessary.

### 3.6. Complications and Adverse Events

Four patients (21%) required revision surgery due to surgical site infection (n = 3) or cerebrospinal fluid leak (n = 1). One patient underwent additional surgery using an anterior approach after experiencing a screw fracture and impaction of a previously placed intervertebral cage. In two patients with persistent neck pain, the hardware was removed.

Since the PCF surgery, four patients (21%) died. The indication for surgery in these cases was neoplastic lesions in three patients and degenerative disease in one patient. There was one case (5%) of perioperative mortality within 30 days of surgery. The patient had prostate cancer with osseous metastases and initially underwent an anterior vertebrectomy of C6 and C7 with anterior plate placement, followed by a posterior fusion from C5 to Th4 10 days later. Although the initial postoperative course was uneventful, with the patient being discharged to rehabilitation 7 days after surgery, the patient died in the rehabilitation facility 15 days later. The exact cause of death remains unknown.

### 3.7. Sensitivity, Specificity, and Accuracy of iCT

In our series, iCT was used in all cases. For upper posterior cervical spine approaches, including C6, lateral mass screws were used, while pedicle screws were the preferred technique for fusion below C6. A total of 151 lateral mass and pedicle screws were evaluated using iCT. Based on the iCT findings, eight screws (5.3%) were repositioned intraoperatively. Correct screw placement was confirmed with a final iCT scan at the end of the surgery. Additionally, all patients underwent a postoperative CT scan, which confirmed the correct placement of 110 screws (72.8%). A minor violation was found in 24 screws (15.9%), a moderate violation in 13 screws (8.6%), and a severe violation in 4 screws (2.6%). No additional surgeries were necessary to reposition the screws.

The sensitivity, specificity, and accuracy of the iCT scans were calculated using the postoperative CT scan as a reference. For correct screw placement, the sensitivity, specificity, and accuracy were 92.7%, 82.9%, and 90.0%, respectively. For minor violations, the values were 70.8%, 91.3%, and 88.1%, respectively. For moderate violations, the sensitivity, specificity, and accuracy were 69.2%, 98.6%, and 96.0%, respectively. For severe screw violations, the sensitivity was 75%, specificity was 100%, and accuracy was 99.3%. Refer to Table 4 and Table 5.

Radiation exposure was documented in 100% of the cases. The mean cumulative air kerma for the intraoperative CT was 43.51 (±42.93) mGy, compared to 66.6 mGy for the postoperative CT scans of the cervical spine. The mean corresponding cumulative dose area product (DAP) was 15.51 (±15.72) Gy cm^2^ in the iCT and 13.15 Gy cm^2^ in the regular postoperative scan.

## 4. Discussion

### 4.1. Clinical Outcome

Our study demonstrated good short- and long-term outcomes, regardless of previous surgical interventions on the cervical spine or the underlying condition. At the 2-year follow-up, 79% of patients reported overall pain improvement, and functional recovery was achieved in up to 88% of cases. These findings are consistent with a systematic review and meta-analysis by Youssef et al. [17], which included data from 1238 patients who underwent posterior spinal fusion. They reported similar favorable outcomes, with significant improvements in pain levels and overall function, as assessed by the Japanese Orthopedic Association (JOA) and modified Japanese Orthopedic Association (mJOA) scores. Similarly, Anderson et al. [18] conducted a systematic review of 11 studies and described comparable results, with functional improvement in 70% to 95% of patients and a significant improvement in JOA scores.

In our series, 15 out of 19 patients had a follow-up of at least 12 months. Among all patients, statistically significant improvements were observed in all evaluated categories, including pain, sensorimotor deficits, and gait disturbance. The 15 patients with longer follow-up showed a trend towards progressively better results over time. Although there was no statistical significance between the two groups due to the small sample size, we would expect the four patients who were not followed for at least 12 months to continue improving rather than deteriorating.

The use of iCT in this study allowed for the immediate identification and direct replacement of eight misplaced screws, resulting in improved screw placement accuracy and potentially avoiding the need for subsequent revision surgeries. Patients who underwent intraoperative screw repositioning demonstrated comparable outcomes to those who did not require screw adjustment, with no significant differences in pain reduction, neurological function, or quality-of-life measures in both short- and long-term follow-up. Based on these findings, it is reasonable to hypothesize that the implementation of iCT technology could lead to better clinical outcomes for patients undergoing spinal instrumentation procedures. However, to thoroughly evaluate the impact of iCT on patient outcomes and to establish its superiority over alternative techniques, such as navigated screw placement, further controlled and prospective studies are necessary.

### 4.2. Surgical Approach

Various surgical approaches to the cervical spine, with or without fusion of the segment, have been described in the literature. Anterior cervical approaches include anterior cervical discectomy with cage only or with an additional plate, as well as corpectomy with fusion. For posterior approaches, laminotomy, laminoplasty, and laminectomy with or without fusion are the most common techniques. In cases of symptomatic cervical disk herniations, anterior cervical discectomy and fusion (ACDF) is a well-established and routine procedure. However, this approach is associated with typical complications such as injury to the recurrent laryngeal nerve and dysphagia [19]. In cervical spondylotic myelopathy (CSM), spondylotic changes and deformities lead to the compression of the spinal cord. The most favorable approach for treating CSM remains controversial [20].

A retrospective study conducted at our institution found that posterior decompression without fusion was an effective treatment option for CSM, significantly relieving symptoms in patients without signs of cervical spinal instability prior to surgery [21]. In contrast, other authors have recommended avoiding stand-alone laminectomy due to the inherent risk of delayed postoperative kyphosis [22]. For patients with pre-existing neck pain, kyphosis, and signs of instability, posterior fusion should generally be the preferred approach [23].

Asher et al. [24] published a multicenter analysis comparing anterior and posterior approaches. They found that patients undergoing posterior fusion were more likely to receive a fusion involving more than three levels compared to the anterior fusion group. While the length of hospital stay was significantly longer in the posterior group, long-term reported outcomes and complication rates were similar between the two approaches [24].

A prospective study [25] involving a total of 264 patients compared anterior discectomy and fusion with posterior decompression (laminoplasty or laminectomy with fusion). In this series, the anterior approach showed favorable results. However, patients receiving posterior decompression were more likely to suffer from multi-level pathologies and were generally older. After adjusting for these confounding factors, similar results were reported for both anterior and posterior approaches [25].

In trauma patients, the choice of surgical approach depends on additional factors such as the presence and volume of herniated disks, the presence of bone fragments, concomitant spondylotic changes narrowing the spinal canal, the presence of uni- or bilateral facet dislocation, and the patient’s neurological status. Finally, it is important to note that posterior fusion of the cervical spine is a more demanding technique, and the surgeon’s training level and experience play a crucial role.

In our current series, 63% of patients were initially treated using an anterior approach. Posterior fusion was the first choice for patients with preoperative instability or malalignment of the cervical spine caused by fractures or tumor growth. Posterior fusion following anterior fusion was performed if symptoms persisted, mainly in cases of malalignment and instability.

### 4.3. Adverse Events and Revision Rate

Other studies have reported complication rates of 11% and 11.6% [23,26]. We attribute the higher perioperative complication rate in our series to the small number of patients included. All patients who underwent revision surgery showed a good outcome with complication-free wound healing. More significant surgical complications, especially those resulting in new persistent neurological deficits or long-term disability, did not occur. Notably, due to the use of iCT, no additional surgery was required at a later stage to correct the alignment of misplaced screws.

### 4.4. Radiological Outcomes and Fusion Rates

At the final radiological follow-up, 92% of patients showed signs of partial or complete fusion (Lenke and Bridwell Grades A to C). This finding is consistent with the fusion rates reported in previous studies, which typically ranged from 89% to 100% [27,28]. Only one patient showed a non-union at their last follow-up, which was 7.33 years after posterior fusion. Clinical outcomes in patients with incomplete fusion (Lenke and Bridwell Grades C and D) were comparable to those with solid fusion.

### 4.5. Intraoperative CT Scan

In comparison to the postoperative CT scan, iCT showed a good sensitivity and specificity. All four screws with severe violation were registered on the iCT scan. Of the 13 screws showing a moderate violation, only one misplaced screw was missed on iCT. A previous study by Nevzati E. et al. [10], which was conducted at our institution, evaluated the accuracy of intraoperative CT scans for pedicle screws in the lumbar and caudal thoracic spine. They reported sensitivity and specificity rates for moderate and severe violations exceeding 86%, which slightly surpasses the sensitivity of our results. We attribute this difference to the smaller diameter of cervical lateral mass and pedicle screws, where differences in resolution between iCT and regular CT scans become more important for detecting screw misplacement.

Some authors argue that lateral mass screw placement does not require intraoperative radiographic control, as they are less likely to be misplaced when using anatomical landmarks [29]. This is consistent with the results in our series, where only one out of the eight intraoperatively corrected screws was a lateral mass screw at the C4 level, while the other seven were transpedicular screws. We routinely performed an iCT scan, as it offered the possibility to further reduce the risk of screw misplacement with a reasonably moderate increase in radiation exposure and only a minor extension of the operation time.

Given the high sensitivity (92.7%), specificity (82.9%), and accuracy (90.0%) reported in this study, it can be argued that a routine postoperative CT scan may be unnecessary, as it offers no significant additional benefits compared to iCT.

It is important to note that the use of an iCT scan is just one method to decrease the number of screw misplacements. Several studies have demonstrated the advantages of using intraoperative navigation [30,31,32,33]. Navigated screw placement is used in many centers and has recently also been established at our institution. Each technique offers distinct advantages and limitations. The angiography suite provides versatility beyond spinal surgery, supporting various interventional procedures. While navigation technology reduces radiation exposure for both patients and surgical staff, it may require larger incisions for reference frame placement, potentially increasing soft tissue dissection compared to intraoperative CT.

At our institution, intraoperative CT remained the standard of care for spinal fusion procedures. However, the recent implementation of navigation technology may reduce the need for routine intraoperative CT scanning in the future, as it independently enables accurate screw placement. The measured mean values for cumulative air kerma and dose area product are slightly lower than the values measured in the postoperative CT scans. However, it is important to note that these measurements are difficult to compare. The iCT scans are performed in a flat panel angiography suite with a rotating C-arm, while the postoperative CT scans use a multidetector computed tomography system. As a result, the actual radiation doses affecting the patient are not accurately represented by the measured values. A study by Jones and Odisio [34] used a phantom to compare the radiation doses when using the two different systems. They found that the mean central axis dose in the multidetector CT system was 41–69% lower compared to the flat panel CT scan. However, the measured noise was much higher in the multidetector CT systems. When noise magnitudes were matched, similar radiation doses were expected between the two systems.

At our institution, cervical fusion procedures are primarily performed via anterior approach without intraoperative CT guidance. The AlluraXper FD20 system maintains cost-effectiveness through its diverse applications beyond spine surgery. The system supports multiple procedures including lumbar spinal fusion, surgical management of intracranial hemorrhages, and vascular neurosurgery. Additionally, the angiography suite serves both neurosurgical and vascular surgery departments. In centers with higher case volumes, the system may achieve cost-effectiveness through spinal fusion procedures alone.

### 4.6. Study Limitations

This retrospective study had several limitations. First, the cohort size was relatively small, which may limit the generalizability of the findings. Our institutional preference for the anterior approach to cervical fusion limits posterior cervical fusion procedures to approximately six cases annually.

Second, due to the retrospective nature of the study, clinical scores such as VAS, NDI, Nurick Scale, and modified Macnab Criteria were not available for all patients, which could have provided a more comprehensive assessment of patient outcomes. Third, among the 13 patients with long-term radiological follow-up, 3 exhibited fusion masses with apparent cracks, suggesting that these may not represent solid fusions. Finally, CT scans were typically performed on symptomatic patients suspected of non-union, which could have led to a biased detection of lower fusion rates.

## 5. Conclusions

The findings of this study suggest that posterior cervical fusion performed in a hybrid OR setting is a safe and effective treatment option for various cervical spinal pathologies. Most patients (88%) achieved good or excellent long-term clinical outcomes, with 84% reporting pain improvement and a fusion rate of 94%. The high sensitivity and specificity of iCT for detecting relevant screw malposition prevented the need for any revision surgery for screw replacement in this complex and surgically demanding area of the spine. These results highlight the potential benefits of using advanced intraoperative imaging technologies in the surgical management of cervical spine disorders. Future prospective studies with larger cohorts are needed to fully evaluate the efficacy of intraoperative CT for spinal fusion procedures, particularly in comparison with alternative techniques such as spinal navigation.

## Figures and Tables

**Figure 1 brainsci-15-00160-f001:**
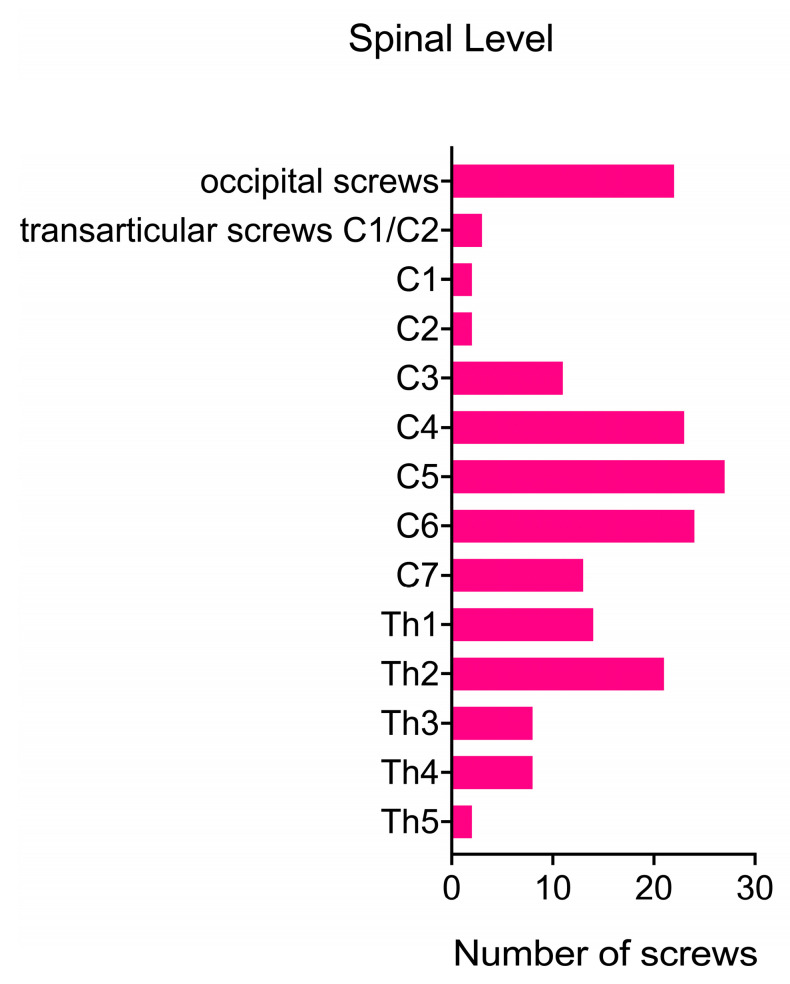
Screw placement: quantity and anatomical distribution.

**Figure 2 brainsci-15-00160-f002:**
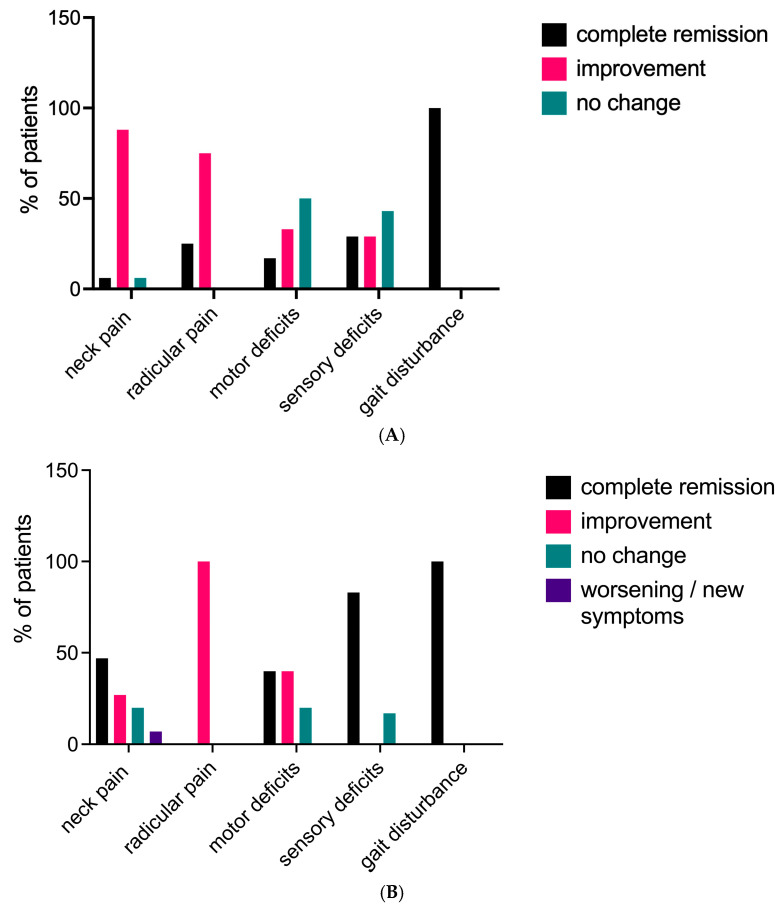
(**A**) Clinical course of symptoms at discharge. (**B**) Evolution of symptoms at final follow-up.

**Figure 3 brainsci-15-00160-f003:**
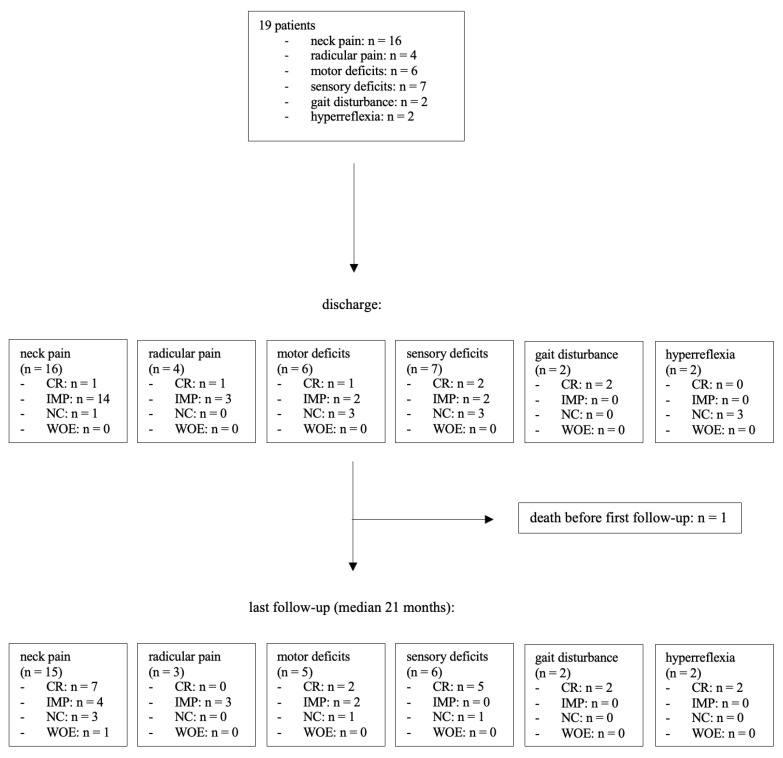
Flowchart demonstrating the clinical course. CR: complete remission, IMP: improvement, NC: no change, WOE: worsening or new symptoms.

**Table 1 brainsci-15-00160-t001:** Patient characteristics.

Mean Age	59 (±11) Years
Sex	
male	12 (63%)
female	7 (37%)
Pathology	
neoplastic	7 (37%)
metastasis	4 (57%)
chordoma	2 (29%)
plasmocytoma	1 (14%)
degenerative	6 (32%)
traumatic	6 (32%)
Fractures	
Proc. condylaris	1
Atlas (Jefferson fracture)	1
Dens (Anderson D’Alonzo Type 2)	2
Hangman’s fracture (Effendi Type 2)	1
incomplete burst (AO A3)	1
hyperextension fracture (AO B2)	1
Preoperative Symptoms	
neck pain	16 (84%)
radicular pain	4 (21%)
sensory deficit	7 (37%)
motor deficit	6 (32%)
gait disturbance	2 (11%)
hyperreflexia	2 (11%)
Preoperative Radiological Findings	
compression of brain stem	1 (5%)
compression of cervical spinal cord	7 (37%)
radiological signs of myelopathy	5 (26%)
compression of nerve root	5 (26%)
cervical spinal instability	4 (21%)
atlanto-occipital subluxation	1 (5%)
spondylolisthesis	3 (16%)
kyphosis	5 (26%)
Mean Number of Segments Undergoing Fusion	4.74
Extent of Surgical Fusion	
cervical spine only	6 (32%)
inclusion of occipito-cervical junction	5 (26%)
inclusion of upper thoracic spine	11 (58%)
inclusion of occipito-cervical junction and upper thoracic spine	1 (5%)

**Table 2 brainsci-15-00160-t002:** Overview of total number and level of screw placement.

Patient	Extent of Fusion	Intraoperatively Repositioned Screws
1	Occipital—C5	
2	Occipital—C5	
3	Occipital—C5	
4	Occipital—C6	
5	Occipital—Th2	C7, right side
6	C1—C2	
7	C3—Th1	C3, left side
8	C3—Th3	Th1, left side
9	C4—C5	
10	C4—C7	
11	C4—C7	
12	C4—Th2	Th1, bilaterally
13	C5—Th1	Th1, left side
14	C5—Th2	
15	C5—Th3	Th2, right side
16	C5—Th4	
17	C5—Th4	
18	C5—Th4	Th4, right side
19	C6—Th4	

**Table 3 brainsci-15-00160-t003:** Clinical and radiological scores.

Neck Disability Index		Odom’s Critera	
0 points	3 (43%)	excellent	4 (57%)
1–5 points	2 (29%)	good	1 (14%)
>5 points	2 (29%)	fair	2 (29%)
		poor	
Nurick Scale		Lenke and Bridwell	
0 points	7	Grade A	5 (38%)
1 point	0	Grade B	4 (31%)
2 points	0	Grade C	3 (23%)
3 points	0	Grade D	1 (8%)
4 points	0		
5 points	0		
Modified Macnab Criteria			
excellent	2 (29%)		
good	3 (43%)		
fair	2 (29%)		
poor	0		

**Table 4 brainsci-15-00160-t004:** Assessment of screw placement—comparison of intraoperative versus postoperative CT.

	Intraoperative CT Scan				
Postoperative CT Scan	No Violation	Minor Violation	Moderate Violation	Severe Violation	Total
No violation	102	8	0	0	110
Minor violation	6	17	1	0	24
Moderate violation	1	3	9	0	13
Severe violation	0	0	1	3	4
Total	109	28	11	3	151

**Table 5 brainsci-15-00160-t005:** Sensitivity, specificity, and accuracy of intraoperative CT.

Extent of Violation	Sensitivity (%)	Specificity (%)	Accuracy (%)
No violation	92.7	82.9	90.0
Minor violation (<1/3 screw diameter)	70.8	91.3	88.1
Moderate violation (>1/3 screw diameter, <1 screw diameter)	69.2	98.6	96.0
Severe violation (>1 screw diameter)	75.0	100	99.3

## Data Availability

The authors will share the data upon reasonable request.

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
