# Peer review of "Clinical and Radiological Outcome of Posterior Cervical Fusion Using Philips AlluraXper FD20 Angiography Suite"

_brainsci, 2025, doi:10.3390/brainsci15020160_

Round 1
Reviewer 1 Report
Comments and Suggestions for Authors
I want to take a moment to thank the editorial board for the opportunity to review this paper by Dolp et al. The authors present their work regarding improving the quality and safety of posterior cervical fusion procedures using a new interventional X-ray system. Overall, the manuscript is well written; however, I have some suggestions for the authors to consider to enhance the quality of the manuscript, which I will mention below. Please bear with me as these points are only there to enhance the quality of the paper and are not criticisms.
1) I performed a Google search about the Philips AlluraXper FD20. According to the manufacturer's site, this is an interventional X-ray system. I wanted to ask the authors if they should consider rewording the term iCT with iX-ray. That may be more appropriate given the system's X-ray base and non-CT base. Let me know if I am making incorrect assumptions.
2) The premise of the study is great in that this interventional system allows surgeons real-time feedback about the appropriateness of the screws placed during the procedure. Given that this was a new system that was introduced in your hospital, please make sure that in your introduction and the methods, you mention the practice that was prevalent before the introduction of the new system and how many adverse events related to screw misplacement were observed before and after. Please also add a paragraph about what prompted your hospital to purchase this system.
3) Point #2 may also bring out the efficacy of the new system in that the total number of misplaced screws may be significantly reduced compared to the prior. Also, mention how were screw placement detected before the iX-ray system was implemented? Were neuromonitoring such as SSEP and MEP performed during the procedure that may give clues? Were portable X-RAYS used that provided information? Readers, especially from hospitals that face the same challenges as yours, may be able to learn from your paper how to proceed with the robust implementation of a new device that provides patients with the highest quality of care.
3) I like your robustness in follow-up and data collection. Please also add the overall impact of the I-XRAY system, providing the number of cervical fusion procedures your hospital performs every year, and this may also be a good time to suggest cost benefits or neurological deficits prevented due to improper screw placement. If you can project yearly patients that may benefit, that would be very impactful.
4) One concern I have is that the study period is 2012-2016, albeit a follow-up study; even with a median follow-up time of 21 months, the data is more than 5 years removed. Can you justify why you waited this long to publish your findings?
5) Also add if any guidelines from spine surgery societies recommend intraoperative CT or XRAY to verify screw placement. I am assuming this is the standard of care.
6) What is the rate of screw misplacement in the last 5 years?. That data may be worth adding to your results. Even documenting that the rates have been kept low due to the device is worth reporting.
Thank you again for this opportunity to review. I look forward to the revised version of the manuscript. Good luck.
Author Response
Reviewer 1
- I performed a Google search about the Philips AlluraXper FD20. According to the manufacturer's site, this is an interventional X-ray system. I wanted to ask the authors if they should consider rewording the term iCT with iX-ray. That may be more appropriate given the system's X-ray base and non-CT base. Let me know if I am making incorrect assumptions.
We thank the reviewer for the observation. The Philips AlluraXper FD20 at our institution primarily serves as an angiography suite for interventional procedures. In vascular neurosurgery, it enables intraoperative angiography to confirm complete aneurysm exclusion after clipping while verifying preserved blood flow. However, in this study, we utilized the system solely for three-dimensional imaging without endovascular intervention. We chose the term "intraoperative CT" (iCT) to distinguish this application from the conventional two-directional X-ray guidance used at our institution prior to 2010. Therefore, we propose maintaining the term "iCT" throughout the manuscript.
- The premise of the study is great in that this interventional system allows surgeons real-time feedback about the appropriateness of the screws placed during the procedure. Given that this was a new system that was introduced in your hospital, please make sure that in your introduction and the methods, you mention the practice that was prevalent before the introduction of the new system and how many adverse events related to screw misplacement were observed before and after. Please also add a paragraph about what prompted your hospital to purchase this system.
Thank you pointing this out. We have added the practice before 2010 to our materials and methods [line 96]. The rational for purchasing the system was also added [line 102].
- Point #2 may also bring out the efficacy of the new system in that the total number of misplaced screws may be significantly reduced compared to the prior. Also, mention how were screw placement detected before the iX-ray system was implemented? Were neuromonitoring such as SSEP and MEP performed during the procedure that may give clues? Were portable X-RAYS used that provided information? Readers, especially from hospitals that face the same challenges as yours, may be able to learn from your paper how to proceed with the robust implementation of a new device that provides patients with the highest quality of care.
Prior to the implementation of iCT, screw placement was verified using conventional portable X-rays during surgery. However, since screw misplacements weren't systematically documented in this earlier period, we cannot provide a direct statistical comparison of misplacement rates before and after iCT implementation. Nevertheless, our study demonstrates that iCT led to the immediate intraoperative detection and correction of 8 misplaced screws that might have gone unnoticed with conventional imaging. This real-time correction capability represents a significant advancement in ensuring accurate screw placement and improving patient safety.
Regarding intraoperative neuromonitoring, we don't routinely use SSEP and MEP for spinal fusion procedures at the Cantonal Hospital Aarau, as noted in our methods section [line 125]. Our transition to iCT was driven by the need for more precise, real-time imaging during surgery. We appreciate this important finding highlighted by the reviewer and believe our experience with implementing this technology and the resulting workflow adaptations could be valuable for other institutions considering similar upgrades to their surgical imaging capabilities.
- I like your robustness in follow-up and data collection. Please also add the overall impact of the I-XRAY system, providing the number of cervical fusion procedures your hospital performs every year, and this may also be a good time to suggest cost benefits or neurological deficits prevented due to improper screw placement. If you can project yearly patients that may benefit, that would be very impactful.
While we acknowledge the relatively low number of posterior cervical spinal fusions in our study, which is addressed in our limitations section [line 417] and primarily results from our preferred anterior approach, we maintain that the AlluraXper FD20 system remains financially viable. As detailed in section 4.5 [line 406], this viability stems from the system's versatility, allowing its use across multiple surgical applications beyond just spinal procedures.
- One concern I have is that the study period is 2012-2016, albeit a follow-up study; even with a median follow-up time of 21 months, the data is more than 5 years removed. Can you justify why you waited this long to publish your findings?
We acknowledge the time gap between the study period (2012-2016) and the current publication. While the data remains relevant for evaluating the implementation and efficacy of the iCT system, we experienced delays in our data collection and analysis process. Despite this administrative delay, the findings provide valuable insights into the integration and utilization of intraoperative imaging technology in spinal surgery, particularly given the extended follow-up period (median 21 months) which allowed us to assess long-term outcomes.
- Also add if any guidelines from spine surgery societies recommend intraoperative CT or XRAY to verify screw placement. I am assuming this is the standard of care.
There is good evidence for the efficacy of iCT for the placement of pedicle screws especially in the thoracic and lumbar spine.
Scarone P, Vincenzo G, Distefano D, et al. Use of the Airo mobile intraoperative CT system versus the O-arm for
transpedicular screw fixation in the thoracic and lumbar spine: a retrospective cohort study of 263 patients. J Neurosurg
Spine. Oct 2018;29(4):397-406. doi:10.3171/2018.1.SPINE17927
Farah K, Coudert P, Graillon T, et al. Prospective Comparative Study in Spine Surgery Between O-Arm and Airo Systems:
Efficacy and Radiation Exposure. World Neurosurg. Oct 2018;118:e175-e184. doi:10.1016/j.wneu.2018.06.148
Fichtner J, Hofmann N, Rienmüller A, Buchmann N, Gempt J, Kirschke J et al. Revision Rate of Misplaced Pedicle Screws
of the Thoracolumbar Spine–Comparison of Three-Dimensional Fluoroscopy Navigation with Freehand Placement: A
Systematic Analysis and Review of the Literature. World Neurosurgery. 2018;109:e24-e32.
While there are currently no official guidelines specifically addressing iCT use in posterior cervical spinal fusion, likely due to its less widespread adoption, the German Society for Orthopedic and Trauma Surgery (DGOU) recommends the use of navigation for pedicle screw placement in C3 to C6 following subaxial cervical spine injuries. This recommendation indirectly supports the utility of advanced imaging technologies like iCT in ensuring accurate screw placement in cervical procedures.
Schleicher P, Kobbe P, Zimmermann V, et al. Treatment of Injuries to the Subaxial Cervical Spine: Recommendations of the Spine Section of the German Society for Orthopaedics and Trauma (DGOU). Global Spine J. 2018 Sep;8(2 Suppl):25S-33S.
- What is the rate of screw misplacement in the last 5 years? That data may be worth adding to your results. Even documenting that the rates have been kept low due to the device is worth reporting.
While we haven't systematically documented screw misplacement rates since the conclusion of our study period, the iCT system has remained in continuous use as part of our standard surgical protocol, as noted in our discussion section [line 389]. This continued reliance on the technology reflects our surgical team's confidence in its utility for ensuring accurate screw placement. A systematic follow-up study examining long-term misplacement rates would indeed be valuable for demonstrating the sustained benefits of iCT implementation.
Reviewer 2 Report
Comments and Suggestions for Authors
The manuscript Clinical and radiological outcome of posterior cervical fusion using philips alluraXper FD20 angiography suite by Armando Dolp et al. investigates the clinical and radiological outcomes of posterior cervical fusion (PCF) performed in a hybrid operating room, focusing on the use of intraoperative computed tomography (iCT) to detect and correct screw misplacement. The objective of the study is to evaluate the effectiveness of iCT in improving screw placement accuracy, minimizing revision surgeries, and ensuring patient safety.
Comments to the Authors:
- While the study provides relevant contributions, the small sample size (19 patients) limits the generalization of the results. It would be interesting to include this limitation in the manuscript.
- The fusion rate reported in the manuscript (92%) is significant, but the manuscript mentions the presence of cracks in fusion masses in three patients. It would be important to explore or comment on whether these findings have any influence on clinical outcomes and how this relates to the overall success rate.
- The comparison of radiation exposure between iCT and postoperative CT scans is briefly mentioned in the manuscript. Including a detailed analysis or references to comparative studies on radiation doses and patient safety would strengthen the discussion.
- The manuscript discusses that iCT allowed the immediate correction of eight screws. Could you please expand on how these corrections influenced long-term clinical outcomes, such as mobility and pain relief? This would add depth to the results.
- Although the manuscript discusses the accuracy of screw placement, the inclusion of charts or tables detailing the classification and correction of positioning violations could improve clarity and presentation of the results.
- The discussion briefly mentions navigated screw placement as an alternative to iCT. Expanding on the advantages, limitations, and cost implications of each approach would provide a more comprehensive evaluation of the clinical utility of iCT.
- Verify if all abbreviations are properly defined from the beginning of the manuscript.
- A table summarizing the key findings, including intraoperative corrections and postoperative outcomes, could enhance the clarity of the manuscript.
- The conclusion could include the potential clinical impact of iCT and outline directions for future research, especially prospective studies with larger samples.
Author Response
Reviewer 2
- While the study provides relevant contributions, the small sample size (19 patients) limits the generalization of the results. It would be interesting to include this limitation in the manuscript.
We fully acknowledge this limitation and have addressed it in multiple sections of the manuscript. Specifically, in section 4.6 (study limitations) [line 417], we discuss both the small sample size and explain its primary reason - our institutional preference for the anterior surgical approach in cervical spine procedures. Furthermore, recognizing the need for more robust evidence, we explicitly call for prospective studies with larger patient cohorts in our conclusions [line 439] to better establish the generalizability of our findings.
- The fusion rate reported in the manuscript (92%) is significant, but the manuscript mentions the presence of cracks in fusion masses in three patients. It would be important to explore or comment on whether these findings have any influence on clinical outcomes and how this relates to the overall success rate.
Thank you for highlighting this important point. We have expanded our discussion of these cases in section 4.4 [line 355] to clarify that the patients with observed cracks in their fusion masses demonstrated clinical outcomes comparable to those without such findings. These observations support our overall fusion success rate of 92%, suggesting that minor radiological findings such as these cracks did not adversely impact the clinical outcomes or the overall effectiveness of the procedure.
- The comparison of radiation exposure between iCT and postoperative CT scans is briefly mentioned in the manuscript. Including a detailed analysis or references to comparative studies on radiation doses and patient safety would strengthen the discussion.
We appreciate this suggestion regarding radiation exposure analysis. In our study, we systematically recorded radiation doses for all patients and attempted to contextualize these measurements against alternative radiological approaches and institutional protocols. However, three major challenges complicate direct comparisons:
- The technical differences between flat panel and multidetector systems make radiation dose comparisons inherently complex. We have addressed this challenge by referencing a comparative study in our discussion [line 399].
- The relatively limited use of iCT in cervical fusion procedures results in a scarcity of comparable studies in the literature.
- The lack of standardization in radiation dose measurement and reporting methodologies across research groups further complicates meaningful comparisons.
- The manuscript discusses that iCT allowed the immediate correction of eight screws. Could you please expand on how these corrections influenced long-term clinical outcomes, such as mobility and pain relief? This would add depth to the results.
Thank you for this important point about long-term outcomes. We have expanded our discussion [line 290] to clarify that patients who underwent intraoperative screw replacement demonstrated comparable clinical outcomes to those who did not require such corrections, both in the immediate post-operative period and during long-term follow-up. This observation suggests that the ability to detect and correct screw placement in real-time not only prevents potential complications but also ensures that these corrections do not compromise the overall clinical success of the procedure in terms of mobility and pain relief.
- Although the manuscript discusses the accuracy of screw placement, the inclusion of charts or tables detailing the classification and correction of positioning violations could improve clarity and presentation of the results.
Thank you for this suggestion regarding data presentation. While we understand the potential value of detailed classification tables, we believe that presenting individual data for all 151 screws might overwhelm readers rather than enhance clarity. Instead, we have provided a comprehensive overview in Table 2 [line 167], which details the extent of fusion for each case and identifies the levels where screw repositioning was required. This format allows readers to efficiently grasp both the scope of the procedures and the distribution of corrections while maintaining clarity in data presentation.
- The discussion briefly mentions navigated screw placement as an alternative to iCT. Expanding on the advantages, limitations, and cost implications of each approach would provide a more comprehensive evaluation of the clinical utility of iCT.
Thank you for this valuable suggestion regarding comparative analysis. We have expanded section [line 382] of our discussion to provide a more comprehensive evaluation of both iCT and navigated screw placement techniques. The revised section now includes a detailed comparison of the advantages and limitations of each approach, including technical considerations, workflow implications, and economic aspects. We believe this expanded discussion better contextualizes our choice of iCT while acknowledging the merits of alternative approaches.
- Verify if all abbreviations are properly defined from the beginning of the manuscript.
Thank you for your suggestion regarding abbreviations. We have carefully reviewed the manuscript and confirmed that all abbreviations are properly defined at their first use. Additionally, for the reader's convenience, we have added a comprehensive list of abbreviations at the beginning of the manuscript [line 31].
- A table summarizing the key findings, including intraoperative corrections and postoperative outcomes, could enhance the clarity of the manuscript.
Thank you for this constructive suggestion. We agree that summarizing the key findings would enhance clarity. In the current manuscript, we have presented this information across several elements:
- Table 2 [line 167] provides a comprehensive list of all intraoperatively corrected screws.
- Figure 3 [line 206] presents a flowchart detailing the clinical outcomes at both discharge and final follow-up.
- Figure 2A and 2B [lines 202 and 204] offer graphic representations of these findings.
However, if you feel a single consolidated table would better serve the readers, we would be happy to create one that combines these key findings.
- The conclusion could include the potential clinical impact of iCT and outline directions for future research, especially prospective studies with larger samples.
Thank you for this valuable suggestion. We have expanded our conclusion section [line 439] to include the potential clinical implications of intraoperative CT imaging and have outlined several directions for future research, particularly emphasizing the need for prospective studies with larger patient cohorts to further validate our findings.
Round 2
Reviewer 1 Report
Comments and Suggestions for Authors
The authors have addressed my concerns and incorporated the suggestions to my satisfaction.